# Degradable and Non-Degradable Chondroitin Sulfate Particles with the Controlled Antibiotic Release for Bacterial Infections

**DOI:** 10.3390/pharmaceutics14081739

**Published:** 2022-08-20

**Authors:** Selin S. Suner, Mehtap Sahiner, Ramesh S. Ayyala, Nurettin Sahiner

**Affiliations:** 1Department of Chemistry & Nanoscience and Technology Research and Application Center, Canakkale Onsekiz Mart University Terzioglu Campus, Canakkale 17100, Turkey; 2Bioengineering Department, Engineering Faculty, Canakkale Onsekiz Mart University Terzioglu Campus, Canakkale 17100, Turkey; 3Department of Ophthalmology, Morsani College of Medicine, University of South Florida, Tampa, FL 33612, USA; 4Department of Chemical and Biomolecular Engineering, University of South Florida, Tampa, FL 33620, USA

**Keywords:** chondroitin sulfate, CS microgels/nanogels, controlled degradation, drug delivery, tobramycin/amikacin, *Pseudomonas* keratitis

## Abstract

Non-degradable, slightly degradable, and completely degradable micro/nanoparticles derived from chondroitin sulfate (CS) were synthesized through crosslinking reactions at 50%, 40%, and 20% mole ratios, respectively. The CS particles with a 20% crosslinking ratio show total degradation within 48 h, whereas 50% CS particles were highly stable for up to 240 h with only 7.0 ± 2.8% weight loss in physiological conditions (pH 7.4, 37 °C). Tobramycin and amikacin antibiotics were encapsulated into non-degradable CS particles with high loading at 250 g/mg for the treatment of corneal bacterial ulcers. The highest release capacity of 92 ± 2% was obtained for CS-Amikacin particles with sustainable and long-term release profiles. The antibacterial effects of CS particles loaded with 2.5 mg of antibiotic continued to render a prolonged release time of 240 h with 24 ± 2 mm inhibition zones against *Pseudomonas aeruginosa*. Furthermore, as a carrier, CS particles significantly improved the compatibility of the antibiotics even at high particle concentrations of 1000 g/mL with a minimum of 71 ± 7% fibroblast cell viability. In summary, the sustainable delivery of antibiotics and long-term treatment of bacterial keratitis were shown to be afforded by the design of tunable degradation ability of CS particles with improved biocompatibility for the encapsulated drugs.

## 1. Introduction

Chondroitin sulfate (CS) is a sulfated glycosaminoglycan (GAGs) comprising N-acetyl galactosamine and glucuronic acid found in proteoglycans in connective tissues [1]. According to the latest scientific literature, there are numerous reports of CS-derived polymeric materials as wound dressing [2], tissue scaffolds [3], coating materials [4], diagnostic devices, biosensors [5], as well as use in hydrogel designs [6]. Furthermore, CSs anti-inflammatory [7], non-immunogenic, non-toxic, and biocompatible abilities [8], as well as its antioxidant [9], anti-atherosclerosis, anti-coagulation, and anti-thrombosis activities make it a promising natural molecule for the design of polymeric systems [10], especially as carrier systems for drugs [11], genes [12], cells, and growth factors [13]. Shi et al. prepared CS-based nanoparticles loaded with cancer drugs synthesized by self-assembly via the redox-sensitive linker of cysteamine [14]. Initially, quercetin and chlorine e6 agents were grafted onto the linear CS backbone, and this CS-based polymer was conjugated with a redox-sensitive linker as cysteamine by the self-assembly process. Another study reported chondroitin sulfate-chitosan nanoparticles for a transdermal drug release system in the treatment of malaria [15]. CS-chitosan nanoparticles were prepared by ionic bonding of oppositely charged chondroitin sulfate and chitosan. Rodriques et al., designed non-crosslinked CS microparticles using a spray-drying technique as a vehicle for antitubercular drugs [16,17]. CS has also been reported as a chelating material with several metal ions such as calcium [18], strontium [19], and magnesium [20] ions in order to enhance their efficiency against osteoarthritis and osteoporosis. The CaCS complex exhibited strong antiosteoporosis properties [18], and the SrCS complex is reported as safe for chondrocytes and osteoblasts and stated that it could enhance collagen production and reduce inflammation [19]. In addition, an MgCS complex was suggested to increase osteoarthritis chondrocyte proliferation and decrease apoptosis [20]. In another study, zinc chondroitin sulfate, ZnCS complex particles were prepared by physical crosslinking by an ion exchange reaction, that is the interaction of Zn(II) ions with negatively charged COO- and SO3- groups on CS. Therefore, ZnCS complex particles have been used as a wound healing material with their perfect antibacterial and anti-inflammatory properties [21]. The physically crosslinked metal–CS particle complexes were afforded many advantages in biomedical applications; however, the ionic interactions between the metal ions and CS polysaccharide could be disrupted by the change in the pH of the medium and these complex particles may not be stable in different aqueous environments in the body.

Chemically crosslinked polymeric particles from natural sources are considered as safe, effective, and stable crosslinked materials that swell in a suitable solvent and can carry various types of active agents to treat diverse diseases [22]. The development of advanced polymeric carrier systems is of considerable interest in material design for clinical treatment because of significant advantages including safety, biocompatibility, biodegradability, less immunogenicity, as well as reducing the toxicity and side effects of the drugs, and enhancing the solubility of the drugs, controlling the release amount, and especially providing the long-term release kinetics and targeted delivery. Moreover, there is no report on the synthesis of chemically crosslinked virgin CS particles in the range of nano to micron size as an antibiotic carrier biomaterial in the treatment of bacterial keratitis for ophthalmic applications. Here, non-degradable, slightly degradable, and completely degradable forms of CS particles were designed to investigate their use as a potential antibiotic delivery device. As reported in our previous study, the degradability of crosslinked carbohydrate-derived particles could be tunable by adjusting the crosslinker ratio in the particle network [23]. Thus, CS particles could also be prepared in the presence of a divinyl sulfone (DVS) crosslinker at different mole ratios of CS repeating units to attain non-degradable, partially degradable, and degradable forms of CS particles. The hydrolytic degradation of CS particles was investigated under physiological conditions in phosphate buffer solution (PBS) at pH 7.4 and 37 °C.

In the treatment of bacterial ulcers on the cornea such as *Pseudomonas* keratitis, the high ophthalmic toxicity and poor pharmacokinetics of the common drugs such as tobramycin and amikacin offer limited use in the clinical application due to the low drug permeability to the epithelial membrane necessitating frequent administration [24]. Sustainable antibiotic delivery by means of natural biocompatible carbohydrate-based polymeric particles to establish prolonged antibiotic delivery at target sites, e.g., into the cornea offers vital infection treatment methods without the need for repeated administration of the toxic drug formulations (e.g., eye drops, ointments). Thus, these types of controlled antibiotic and antifungal drug delivery systems derived from natural polymers can readily attain long-term antibacterial and antifungal effects and afford low systemic toxicity [25]. Recently, promising polymeric carriers have been designed and developed in the treatment of bacterial ulcers on the cornea [26]. Gebreel et al. reported the use of an antibiotic (norfloxacin)-loaded polylactic-co-glycolic acid particles as a carrier in the treatment of infection caused by *Pseudomonas aeruginosa* and demonstrated that this carrier system can inhibit and eradicate the microorganism on the eye [24]. In another study, a tobramycin-carrying lipid nanoparticle complex with hexadecyl phosphate and stearic acid was prepared as an antibacterial agent for the treatment of ocular infections to improve absorption of the antibiotic [27]. This tobramycin carrier system shows preocular retention over one hour and resulted in improved drug bioavailability in the humor. In addition to the complications associated with the drug toxicity and permeability problems, the other drawbacks, e.g., quick drainage of the drug formulations from the corneal system in direct topical applications, can be impeded by the use of antibiotics carried derived from indigenous polymers that exist in the eye. Therefore, the polysaccharides such as CS that innately exists in the extracellular matrix of the connective tissue, cartilage, tendon, and cornea can be employed as antibiotic delivery materials. Furthermore, CS, a well-known mucoadhesive and bio-adhesive biopolymer that reported to render long residence times on the corneal layer [28,29]. Abdullah et al., reported that chondroitin sulfate–chitosan nanoparticles as a drug carrier with high retention and penetration capabilities with improved permeation of the drug in the ocular system [30]. Consequently, in this study, it was hypothesized that CS-based particles can be readily prepared by chemical crosslinking with controllable degradation and used as antibiotic carriers and delivery systems to establish a regulated, extended, and effective antibiotic release kinetic to surmount the disadvantages of the direct drug therapy applications in the treatment of *Pseudomonas aeruginosa*-dependent infections. In this study, tobramycin and amikacin antibiotics were encapsulated in non-degradable CS particles and their release kinetics under physiological conditions (pH 7.4 and 37 °C) and their antibacterial activities against *Pseudomonas aeruginosa* infection, which causes persistent corneal ulcers, were determined up to a 240-h incubation period. Furthermore, the biocompatibility of bare and drug-loaded CS particles was also investigated by employing the blood compatibility tests of hemolysis, blood clotting assays and, cytotoxicity analysis against L929 fibroblasts.

## 2. Materials and Methods

### 2.1. Materials

Chondroitin sulfate A sodium salt (CS, ≥98%, Average MW 10,000–30,000, Biosynth carbosynth, Compton, UK), divinyl sulfone (DVS, 97%, Merck, Darmstadt, Germany), dioctyl sulfosuccinate sodium salt (AOT, 96%, Acros Organics, Geel, Belgium), 2,4-trimethylpentane (isooctane, ≥99.5%, Isolab, Unterfranken, Germany), and acetone (99%, BRK, Istanbul, Turkey) were used for synthesis of CS particles and were used as received. Tobramycin (from local vender, Deva Holding, Istanbul, Turkey) and amikacin hydrate (≥96.5%, Sigma Aldrich, Saint Louis, MO, USA) antibiotics and trichloroacetic acid (99%, Carlo Erba, Val-de-Reuil, France) were purchased and used as received. Nutrient agar (NA, Condolab, Madrid, Spain) and nutrient broth (NB, Merck, Darmstadt, Germany) were used as bacterial growth medium and *Pseudomonas aeruginosa* ATCC 10145 (KWIK-STIK, Microbiologics, St. Cloud, MN, USA) Gram-negative bacteria was used as received. The L929 fibroblast cells (Mouse C3/An connective tissue) were obtained from the SAP Institute, Ankara, Turkey. Trypsin (0.25%, EDTA 0.02% in PBS), Dulbecco’s Modified Eagle’s Medium (DMEM, with 4.5 g/L glucose, 3.7 g/L sodium pyruvate, L-Glutamine 0.5 g/mL), fetal bovine serum (FBS, heat inactivated), and penicillin/streptomycin (10,000 U/mL penicillin, 10 mg/mL streptomycin) were purchased from Panbiotech, Aidenbach, Germany. Dimethyl sulfoxide (DMSO, 99.9%, Carlo Erba, France) and 3-(4,5-dimethylthiazol-2-yl)-2,5-diphenyltetrazolium bromide (MTT agent, 98%, neofroxx, Einhausen, Germany) were purchased and used as received. Ultra-pure deionized water with resistivity of 18.2 M·Ω·cm was obtained from a Millipore Direct-Q 3 UV water purification system (Merck Darmstadt, Germany) and used for the preparation of all aqueous solutions.

The viscosity average molecular weight (M_v_) of CS was determined by using Ubbelohde viscosimeter at room temperature. Briefly, 10 mg/mL concentration of CS solution was prepared in 0.2 M NaCl. Then, the intrinsic viscosity (dL/g) of the CS solution was evaluated by using the following Equation (1).
η_red_ = (t − t_0_)/t_0c_ = (η − η_0_)/η_0c_(1)
where η_red_, η, and η_0_ are the reduced viscosity and intrinsic viscosity of CS solution and the viscosity of the solvent, respectively, and t and t_0_ are the flow time of the CS solution and solvent, respectively [31,32].

The M_v_ of CS was determined by the using Mark–Houwink–Sakurada equation as the following Equation (2).
[η] = K (M_v_)^α^(2)

The parameters of K and ɑ are the constant value for CS-solvent combination at a certain temperature and these values were accepted as K = 5 × 10^−5^ mL/g and ɑ = 1.1 according to the previously reports of Farajdo et al. [33]. M_v_ is the viscosity average MW of CS and the value was determined as approximately 20 × 10^3^ Da.

### 2.2. Preparation of CS Particles

To synthesize CS particles, 0.3 g of CS was dissolved in 10 mL of 0.2 M NaOH solution, and 1 mL of this solution was dispersed in 30 mL 0.2 M AOT/isooctane solution to obtain inverse microemulsion under vigorous stirring at 1000 rpm. Mixing continued under the same conditions for 1 h to obtain a clear CS solution in the inverse micelles. Next, the crosslinker, DVS at 50, 40, and 20 mol % relative to the CS repeating unit, was added to the emulsion media under continuous vortex for dispersion and mixing continued at 1000 rpm for 1 h more at room temperature. The CS particles were precipitated in an excess amount of acetone. Then, the particles were washed with acetone three times by centrifugation at 10,000 rpm for 10 min to remove unreacted chemicals and surfactants. The obtained CS particles were dried with a heat gun and kept in a closed container for further use.

### 2.3. Characterization of CS Particles

Optical light microscope (Olympus, BX53, Tokyo, Japan) and scanning electron microscope (SEM, SU70, Hitachi, Japan) were used to visualize the shape and size of the CS particles. For SEM analysis, dry CS particles were covered with palladium/gold to a few μm under a vacuum for 10 s. The elemental composition of CS particles was determined with an energy dispersive spectrometry (EDS) detector attached to SEM (SU70, Hitachi, Japan). For size distribution analysis, CS-based particles were suspended in DI water at 1 mg/mL concentration and the hydrodynamic average diameter of the CS-based particles were measured by dynamic light scattering (DLS, 90 plus, Brookhaven Instrument Corp., Holtsville, NY, USA) with 35 mW solid state laser detector at an operating wavelength of 658 nm. The average values are given with standard deviations. Fourier transform infrared (FT-IR) spectra of CS-based materials were recorded in the frequency range of 4000 to 650 cm^−1^ with 4 cm^−1^ resolutions by using an FT-IR spectrophotometer (Perkin-Elmer, Spectrum 100, Akron, OH, USA).

### 2.4. Degradation of CS Particles

Degradation capability of CS particles prepared at three crosslinker ratios of 50, 40, and 20 mol % relative to the CS repeating unit was investigated under physiologic conditions in 0.01 M PBS at pH 7.4 and 37 °C. In short, 30 mg of CS particles were suspended in 10 mL of PBS. These suspensions were placed in a water bath adjusted to 37 °C under 300 rpm mixing rate for certain times of 24 h, 48 h, 72 h, and 240 h. Then, the amount of CS particles in the PBS was precipitated by using a centrifuge at 10,000 rpm for 10 min and dried at 50 °C in an oven. The gravimetric degradation was evaluated as weight loss % of CS particles by using Equation (3).
Weight loss% = (M_0_ − M_t_)/M_0_) × 100(3)
where, M_0_ gives the weight of the CS particles initially and M_t_ shows the weight of the CS particles at time t, which was 24 h, 48 h, 72 h, and 240 h.

### 2.5. Preparation of Antibiotic-Loaded CS Particles, CS-Tobramycin, and CS-Amikacin Particles

Antibiotic-loaded CS-Tobramycin and CS-Amikacin particles were prepared by encapsulation process using DVS crosslinker. In short, 30 mg/mL concentration CS solution was prepared in 10 mL of 0.2 M NaOH solution. As a drug solution, 100 mg/mL concentration of Tobramycin or Amikacin drug was dissolved in 1 mL of DI water. These two solutions were mixed for 2 min. Then, 1.1 mL of the drug:CS solution was dispersed in 30 mL 0.2 M AOT/isooctane solution under vigorous stirring at 1000 rpm for 10 min. Then, the crosslinker DVS at 50, 40, and 20 mol % relative to the CS repeating unit, was added to the emulsion medium under continuous vortex for dispersion and mixing continued at 1000 rpm for 1 more h at room temperature. Next, the prepared CS-Tobramycin and CS-Amikacin particles were removed from the reaction medium and precipitated in excess acetone. Then, the particles were washed with acetone three times by centrifugation at 10,000 rpm for 10 min to remove unreacted chemicals and surfactant. Finally, the obtained drug-loaded CS-Tobramycin and CS-Amikacin particles were dried with a heat gun and kept in a closed container for further use.

### 2.6. Drug Release Studies

Drug release studies from 20 mg CS-Tobramycin and CS-Amikacin particles were investigated by dispersing them in 1 mL of PBS at pH 7.4 and transferring this to a dialysis membrane (MW cut off 12 kDA). The membrane containing the CS-tobramycin/amikacin particles was placed into 40 mL of PBS solution (pH 7.4) at 37 °C in a shaker bath. The amounts released into the PBS solution were evaluated by using high-performance liquid chromatography (HPLC, Thermo Ultimate 3000, Germering, Germany) with a refractive index (RI) detector according to the previously proposed procedure for tobramycin and amikacin [34]. Thermo Acclaim 120 C18 column (3 μm, 75 mm, 120 Å, reversed phase, Thermo Scientific, Sunnyvale, CA, USA) was used as a stationary phase at 30 °C column temperature. Mobile phase was prepared as 90:10 (methanol:water, *v*/*v*) mixture adjusted at pH 2 with trichloroacetic acid. The drug solutions were eluted isocratically in mobile phase with 1.0 mL/min flow rate for 10 min and retention times of tobramycin and amikacin drugs were determined at 4.1 and 3.5 min, respectively. The amount of released drug was calculated against the previously determined drug calibration curve in PBS at the same conditions. As the drug release was constant, the PBS solution was discarded, replenished with 40 mL of fresh PBS, and the release amount of drug was measured. The result is reported as cumulative release amounts. The experiments were repeated three times and presented with standard deviations.

### 2.7. Antibacterial Activity Test of CS-Based Particles

Antibacterial activity of tobramycin and amikacin solutions and drug-loaded CS-Amikacin and CS-Tobramycin particles were investigated by using the disc diffusion method against *Pseudomonas aeruginosa* ATCC 10145. To determine the antibacterial activity of drugs as a control group, 20 μL of drug solutions at five different concentrations in physiologic serum, 50, 20, 10, 5, and 2 mg/mL, were treated with *P. aeruginosa* at different incubation times. Separately, 50 mg/mL concentration of drug-loaded CS particle suspension was prepared in physiologic serum and sterilized by UV irradiation for 2 min. Then, 0.1 mL of 10^7^ CFU/mL bacteria suspension in nutrient broth was inoculated on the nutrient agar plates. Immediately, 9 mm sterile discs were placed on the center of the plate. Then, 50 μL of drug-loaded CS particle suspension was dropped on the sterile discs. Next, the plates were incubated at 37 °C for different incubation times. After the incubation, the inhibition zone (mm) was determined as the diameter of the clear zone.

In addition, minimum inhibition concentration (MIC) and minimum bactericidal concentration (MBC) values of the CS-Amikacin and CS-Tobramycin particles were also determined against *P. aeruginosa* by using microtiter broth dilution method. Briefly, 100 μL of nutrient broth as a liquid growth medium was placed into the each well on a 96-well plate and then 100 μL of 50 mg/mL concentration of drug-loaded CS particle suspension was added to the first well and diluted in a sequence by two-fold with the existing medium to prepare from 25 to 0.046 mg/mL concentrations. Then, 10 μL of 10^7^ CFU/mL bacteria suspension in nutrient broth was added to the each well and the plate was incubated at 37 °C for 24 h. The lowest concentration of the drug-loaded CS particle with no visible growing depends on the transparency accepted as the MIC value. The medium for all the transparent wells was inoculated on nutrient agar as a solid medium and incubated at 37 °C for 24 h. The lowest concentration of the drug-loaded CS particle with no growing was accepted as MBC value.

### 2.8. Blood Compatibility with Hemolysis and Blood Clotting Assays

Hemolysis and blood clotting assays were performed to investigate the hemocompatibility of CS-based particles in accordance to the method reported by Sahiner et al. [35]. Human blood was obtained from healthy volunteers and approved by the Clinical Research Ethics Committee of Canakkale Onsekiz Mart University (2011-KAEK-27/2022) and placed into tubes containing EDTA. Before the analysis, all solutions were preheated to 37 °C.

For the hemolysis assay, diluted blood was prepared by using 1:1.25 (*v:v*) ratio of blood:0.9% aqueous NaCl solution and 200 μL of the diluted blood was interacted with CS-based particle suspensions in 10 mL of 0.9% saline solution at 100, 250, 500, and 1000 μg/mL concentrations in a water bath at 37 °C for 1 h. In the separation tubes, 200 μL of the diluted blood was added into 10 mL of 0.9% aqueous NaCl solution with DI water as a negative and positive control, respectively. Then, the tubes were centrifuged at 100 g for five minutes and the absorbance values for the supernatants were measured at 542 nm with UV–Vis spectroscopy (T80+ UV/VIS spectrometer, PG Instrument Ltd., Leicestershire, UK). The hemolysis ratio of the CS-based particles was evaluated using Equation (4).
Hemolysis ratio% = (A_material_ − A_negative_)/A_positive_ − A_negative_) × 100(4)
where A_material_ is the absorbance value of the blood solution interacted with materials in 0.9% aqueous NaCl solution. A_negative_ and A_positive_ are the absorbance values of the blood solution without materials in 0.9% aqueous NaCl solution and in DI water, respectively. All assays were carried out in triplicate and the results are given with standard deviations.

For the blood clotting assay, 80 μL of 0.2 M CaCl_2_ aqueous solution was mixed with 1 mL of blood containing EDTA and immediately 270 μL of this blood was covered with 1, 2.5, 5, and 10 mg of the CS-based particles placed into the tubes. After 10 min, 10 mL of DI water was slowly added into the tubes and centrifuged at 100 g for 1 min. Then, 10 mL of supernatant solution containing non-clotting blood was taken from the tube and diluted with 40 mL of DI water. In the separation tube, 250 μL of the blood containing EDTA was dispersed in 50 mL of DI water as a control. The blood solution was incubated at 37.5 °C in a water bath for 1 h and then, the absorbance value of the supernatant was measured at 542 nm by using UV–Vis spectroscopy. The blood clotting index of the CS-based particles was evaluated from Equation (5).
Blood clotting index = (A_material_ − A_control_) × 100(5)
where A_material_ is the absorbance value of the blood solution interacted with the CS-based particles and A_control_ is the absorbance value of the blood solution without the CS-based particles as a control. All assays were carried out in triplicate and the results are given with standard deviations.

### 2.9. Cytotoxicity of CS-Based Particles by MTT Assay

For cytotoxicity analysis, L929 fibroblast cells were cultured under 5% CO_2_ atmosphere in an air-humidified incubator in DMEM supplemented with 10% FBS and 100 U/mL penicillin–streptomycin at up to 80% confluency. The cell viability % of the fibroblasts in the presence of study materials was analyzed by MTT colorimetric assay, which measures the healthy cells that form formazan crystals via cleavage of the tetrazolium ring of the MTT agent [36]. The L929 fibroblast cells at a density of 5 × 10^5^ cells per mL were suspended in 10 mL of culture media and 100 μL of this cell suspension was seeded onto wells on a 96-well plate. The plate was incubated at 37 °C under 5% CO_2_ in an air-humidified incubator for 24 h. After adherence of the fibroblasts, the culture medium was removed from the wells and 100 μL of drug solution or CS-based particle suspension in culture medium at different concentrations between 50 and 1000 μg/mL was added to the cells. The plate was incubated at 37 °C under 5% CO_2_ in an air-humidified incubator for 24 h more. Only culture medium was used as a control group accepted as 100% viability. At the end of the incubation, the medium was removed from the wells, the cells were washed with PBS at pH 7.4, and 100 μL 0.25 mg/mL concentration of MTT agent diluted in culture medium was added to each well [37]. The plate was incubated at room temperature in the dark for 2 h. Then, MTT solution was removed from the wells and 200 μL of DMSO was placed into the wells to dissolve the formazan crystals. The absorbance of the wells was measured at 590 nm with a plate reader (HEALES, MB-530, Shenzhen, China). The absorbance value of control group was accepted as 100% viability and the decrease in cell viability % was estimated based on the absorbance values of the wells treated with study material by following Equation (6).
Cell viability% = (A_material_/A_control_) × 100(6)
where A_material_ is the absorbance value of the cells interacting with materials and A_control_ is the absorbance value of the untreated cells as control. All assays were carried out in triplicate and the results are given with standard deviation.

### 2.10. Statistical Analysis

Statistical analysis was done using GraphPad Prism 9.1.0. software (GraphPad Software Inc., San Diego, CA, USA) for comparison between the control group and each concentration. A p value less than 0.05 was considered as statistical significance.

## 3. Results and Discussions

In crosslinked polysaccharide particle preparation, DVS is commonly used, and depending on the functionality and hydrophilicity of the polysaccharide, the extent of DVS used in the crosslinked network formation can affect its hydrolytic degradability. In this study, CS was crosslinked with DVS at three different ratios, 50%, 40%, and 20% mole ratios of CS repeating units to prepare non-degradable, slightly degradable, and degradable CS particles, respectively, as illustrated in Figure 1a. To synthesize these particles, hydroxyl groups of CS were reacted with vinyl groups of DVS [23]. The ratio (degree) of crosslinker used in CS particle formation determines the degradability of the CS particles [35]. Optical microscope images of the CS particles (prepared with three different crosslinker ratios, 50%, 40%, and 20% mole ratios of CS repeating units) in an aqueous solution are demonstrated in Figure 1b. The CS particles containing a high ratio of crosslinker, e.g., a 50% mole ratio of DVS containing CS particles (based on the repeating unit of CS) were slightly swollen in DI water, and the crosslinked network did not degrade for a long time (up to 72 h). As the crosslinking ratio was reduced, e.g., to 40%, the CS particles were much more swollen than the 50% crosslinked CS particles, and relatively degradable CS particles were realized. Finally, CS particles containing a lower ratio of crosslinker, e.g., 20% DVS crosslinked CS particle, started to degrade within 5 min in an aqueous environment. Therefore, the 20% mole ratio of DVS is the limit for CS particle synthesis and the DVS extent that is <20% ratio results in no CS particle formation. Thus, the CS particle network with a low crosslinking ratio of 20% was extremely water swollen in an aqueous solution and could readily break down the crosslinked networks within a few minutes.

The degradability of crosslinked polymeric particles is the most important property for in vivo applications to avoid the accumulation of these particles in the body for safe use. Therefore, the degradation ability of CS particles was examined by changing the crosslinker ratio in the particle network with 50%, 40%, and 20% mole ratios. The weight loss % of CS particles with 50%, 40%, and 20% mole ratios after up to 240 h in physiological conditions at pH 7.4 and 37 °C are demonstrated in Figure 1c. The corresponding optical microscope images were also given as Appendix A. As can be seen, no degradation was obtained for CS particles at 50% up to 72 h and they were slightly degraded with 7.0 ± 2.8% weight loss amount at 240 h. The optical microscope images of 50% CS particles supported these degradation results. It is clearly seen that non-degradable CS particles crosslinked at the 50% mole ratio are more stable within 72 h and some large size particles were degraded after 240 h. The 40% CS particles had a slow degradation from 7.5 ± 2.1% to 52.5 ± 3.5% between 24 h and 240 h. The optical microscope images of CS particles at 40% demonstrated some degradation up to 48 h, but more than half of the micron-sized CS particles were degraded at 240 h. Therefore, it could be said that 40% CS particles are a slightly degradable material in comparison to non-degradable 50% CS particles. Furthermore, CS particles were quickly degraded almost totally within 48 h with a low crosslinker ratio in 20% CS particles. All these results indicate that the degradability of CS particles can be controlled by regulating the crosslinker ratio in the particle network. A lower crosslinker ratio triggered the higher swelling ability of the particles and provided a slight degradability for the particles. The hydrodynamic size distribution of CS particles was given with DLS measurements as presented in Figure 1d. The average size distribution of non-degradable 50% CS particles was measured as 1079 ± 30 nm with 0.5–5 μm size range.

The particle shape, size, and surface morphology of the dry CS particles were assessed by SEM images and their size distribution was determined using ImageJ software from the SEM images of CS particles given in Figure 2a–c.

As can be seen, the dry CS particles, crosslinked at 50%, had a smooth surface, and a distinct morphology with spherical shapes in the range of 0.5 to 5 μm. On the other hand, the dry form of CS particles crosslinked at 40% had almost spherical shapes with fragmented particle structures in the 5–40 μm size range. On the other hand, the surface structure of the dry CS particles crosslinked with a 20% mole ratio is rough with almost spherical shapes with particle sizes ranging from 5 to 50 μm. The size distribution range for 50% crosslinked CS particles were found to decrease almost ten-fold and five-fold with respect to 40% and 20% crosslinked CS particles, respectively. These images clearly indicate that the perfect shape of CS particles could be synthesized by using a minimum 50% ratio of DVS crosslinker. According to the SEM-EDS analysis, the wt% of elements in CS particles crosslinked at 50% and 40% are demonstrated in Appendix A. The sulfur content of the CS particles at 50% and 40% was found as 13.5% and 6.5%, owing to the SEM-EDS analysis, but theoretically, these values could be 12.3% and 11.7%, respectively. This sulfur content, higher than 5%, came from the existence of more DVS crosslinker in the particle network of 50% CS particles. The highly crosslinked CS particle at 50% is more stable in an aqueous solution while relatively less stable and the 40% crosslinked CS particle could lose some of the un-crosslinked CS and DVS in the washing process of the particles due to the water-soluble nature of the CS polymer.

Figure 2d shows the FT-IR spectra for linear CS and its particle that are crosslinked at three ratios: 50%, 40%, and 20%. The spectra of CS particles at all crosslinker ratios were observed to contain the specific bands of linear CS as reported by a previous study [38]. Briefly, the broad band from 3600 to 3000 cm^−1^ related to OH and NH stretching, CH_2_ stretching at 2934 cm^−1^, amide I band at 1606 cm^−1^, carbon–hydrogen vibrations at 1408 and 1375 cm^−1^, S=O stretching at 1223 cm^−1^, carbon–carbon vibration at 1040 cm^−1^, and C-O-S stretching at 849 cm^−1^ were determined for all CS particles coming from the linear CS chains in the particle network. The CS particles crosslinked at a 20–50% mole ratio had similar vibrational bands. The broad and strong peak at 1040 cm^−1^ in the spectra of CS and low crosslinked CS turn into sharp peaks at 1050 cm^−1^ with particle formation, as can be seen in the spectrum of highly crosslinked CS particles. The shoulder peak at 1110 cm^−1^ and this sharp peak at 1050 cm^−1^ is attributed to the presence of S=O stretching vibrations from sulfoxide groups of the crosslinker, DVS. The width of the peak at 1050 cm^−1^ was significantly decreased with the increase in the ratio of crosslinker (DVS) for CS particles and while the peak intensity at 1110 cm^−1^ was increased with the increase in the ration of the crosslinker. These results indicated that the crosslinker content of CS particles was increased by changing the DVS ratios from 20% to 50%.

*Pseudomonas aeruginosa* corneal ulcers are a significant severe infection compared with the other bacterial ulcers on the cornea [39]. Tobramycin and amikacin are aminoglycoside antibiotics against a broad antibacterial spectrum, but these drugs are generally used as eye drops for the treatment of infections caused by *P. aeruginosa* because they have a higher activity than other antibiotics such as gentamicin in suppressing *Pseudomonas* keratitis on the eye [40]. Side effects such as tearing, swelling of the eye, itching, stinging, and burning of the eye, temporary blurred vision, and nephrotoxicity and ototoxicity limit the direct use of these drugs [41]. Furthermore, the high administration frequency of these drugs, every 1 h, also makes them difficult to use for some patients. Recently, the design of polymeric carriers was established to avoid these toxicity and high frequency dosing problems. The prepared CS particles were utilized as a drug carrying vehicle for tobramycin and amikacin antibiotics for the treatment of corneal *P. aeruginosa* ulcers. The encapsulation technique was used to load these antibiotics into the CS particle network because of multiple advantages that CS can render such as decreasing drug toxicity and side effects, enhancing loading and release capacity, prolonged release kinetics, and so on. Therefore, the encapsulation technique was preferred in this study as a loading process instead of the adsorption or conjugation of the drug as loading techniques. In the synthesis of crosslinked polymeric systems as a carrier material, drug molecules readily embedded inside the particles during the crosslinking reaction [42]. The hydrodynamic size distribution and the polydispersity index values of bare and drug-loaded CS particles crosslinked at a 50% mole ratio were determined and provided in Table 1.

The average size of bare CS particles was slightly decreased upon drug loading, for example, from 1079 ± 30 nm to 830 ± 25 nm for CS-Tobramycin particles and to 776 ± 57 nm for CS-Amikacin particles. The crosslinking reaction of CS polymer with 50% DVS was done in an emulsion medium in the presence of drug molecules. The presence of both drug molecules can affect the size of the CS particles because of the interaction of functional groups of the drug molecules with the functional groups of CS moieties and the crosslinker. The PDI values of the particles were measured between 0.273 and 0.359. These low PDI values of CS-based particles are almost within the acceptable limit for drug delivery applications as the PDI values for this purpose for liposome and nanoliposome formulations are suggested as ≤0.3 [43]. Therefore, these results for drug formulations are injectable size ranges with submicron average particle sizes in an aqueous medium providing promising application potential.

Drug release amount and release capacity of CS-Tobramycin and CS-Amikacin particles crosslinked at 50%, 40%, and 20% mole ratios are given in Table 2.

There has been growing interest in the design of antibacterial polymeric systems for fighting resistant infections in different places of the body [44]. Three common methods have been employed to generate antibacterial polymeric systems; polymeric structures could be designed by directly using antibacterial molecules, or by coating/grafting with antibacterial agents, or loading well-known antibiotics into polymeric structures [45,46]. Because of the well-known high antibacterial ability and controllable loading and release capacities with low toxicity, well-known antibiotics can be loaded into the polymeric networks through different drug loading techniques to treat infections. The encapsulation technique was used as a drug loading process, which is known as drug entrapment during the crosslinking reaction of CS. In the preparation of drug-loaded CS particles, 10 mg of drug was used to be encapsulated into 40 mg of CS particles to attain CS-Tobramycin/Amikacin particles.

Therefore, the drug-loaded amount of CS-tobramycin/amikacin particles for each formulation is 250 μg/mg assuming the entrapment efficiency as 100%. The high amounts of tobramycin and amikacin, approximately 250 μg/mg, were loaded into the CS particle network by the encapsulation technique, and the release profiles of these drugs from CS-Tobramycin and CS-Amikacin particles crosslinked at 50, 40, and 20% mole ratios under physiological conditions at pH 7.4 and 37.5 °C are shown in Figure 3a,b, respectively.

A high amount of drug as 192 ± 3 μg/mg of tobramycin or 214 ± 2 μg/mg amikacin was quicky burst from the low crosslinked CS-Tobramycin/Amikacin particles with 20% crosslinking, for 24 h, because of the fast degradable nature of CS particles crosslinked at 20%. Similarly, 215 ± 8 μg/mg of tobramycin or 242 ± 4 μg/mg amikacin was released from CS-Tobramycin and CS-Amikacin particles 40%, respectively, within 72 h, as seen in Figure 3. Moreover, sustainable release kinetics were observed within 150 h for both drugs and a maximum 200 ± 2 μg/mg of tobramycin and 228 μg/mg of amikacin were released cumulatively from the drug-loaded CS particles crosslinked at 50% over 240 h. It could be said that low crosslinked CS-Tobramycin/Amikacin particles at 20% crosslinking could not encapsulate or load all the drugs during synthesis because of the low crosslinking yield. Therefore, the highest release profile was obtained for CS-Tobramycin/Amikacin particles prepared at 40% crosslinking due to the faster and greater swelling ability of these particles in comparison to the 50% crosslinked CS particles. These results indicate that the release capacity of 50% crosslinked CS-Tobramycin particles was 80 ± 0.8%, but CS-Amikacin particles could release 91 ± 2% of the loaded drug within 240 h and high crosslinked CS particles 50% have been used as promising materials in the treatment of *Pseudomonas* keratitis because of the high concentration and long-term antibiotic release. In the literature, tobramycin/amikacin carriers such as alginate/chitosan particles [47], polyethylene glycol-based hydrogel [48], and liposomal systems [49] were designed to prevent infections caused by *P. aeruginosa*. Deacon et al. reported that alginate/chitosan-based particles were loaded with 92 ± 18 μg/mg tobramycin and 80% of the loaded tobramycin could be delivered from the polymeric network within 48 h [47]. Postic et al. stated that polyethylene glycol-based hydrogels could release approximately 40 μg/mL of amikacin in 5 h [48]. In a study, amikacin-loaded nanoparticles from Eudragit RS 100/Eudragit RL 100 polymer composition were used in the bacterial treatment of ocular infection. The release profiles from the nanocarrier system were finished within 12 h with an approximate 90.8% release capacity [50]. Therefore, in comparison to these carrier systems, CS-Tobramycin/Amikacin particles reported here are promising biomaterials due to the higher antibiotic loading ability and longer time antibiotic release capacities.

For bacterial ulcer treatment, almost a 300 μg/100 μL concentration of drug solution (two drops) is generally recommended every 1 h for severe infections for 1 day, followed by continuous applications every 4–8 h per day [51]. Almost similar drug doses could be administered with only one administration of 2.5 mg drug-loaded CS particles crosslinked at 50%, which could be released as a total of 500 μg tobramycin or 570 μg amikacin within 240 h. Thus, 50 μL of 50 mg/mL (2.5 mg) drug-loaded CS particles 50% were used for further antibacterial activity tests.

Antibacterial activities of CS-Tobramycin particles 50% and CS-Amikacin particles 50% were investigated by the disc diffusion assay with 6 h to 240 h incubation times against Pseudomonas aeruginosa and tobramycin and amikacin drugs alone were used as a control. Inhibition zone diameters for 20 μL of tobramycin and amikacin drugs between 2 and 50 mg/mL, which are equal to 40–1000 μg drugs, and 50 μL of 50 mg/mL CS-Tobramycin and CS-Amikacin particles 50% corresponding to 2.5 mg drug-loaded CS particles loaded are illustrated in Figure 4.

According to the release study, nearly 277 ± 13 μg of tobramycin and 322 ± 14 μg of amikacin was released from 2.5 mg CS-Tobramycin and CS-Amikacin particles within 24 h, respectively. The inhibition zones for 2.5 mg CS-Tobramycin and CS-Amikacin particles after 24 h incubation was 24 ± 2 mm and 28 ± 0 mm, respectively, which are almost similar to the inhibition zones for 200 μg drug solutions determined as 25 ± 1 mm. Antibacterial effects of CS-Tobramycin and CS-Amikacin particles remained the same for up to 240 h incubation time against *P. aeruginosa* because of sustainable release during the long-term period of 240 h.

Furthermore, the inhibition zone diameter, the minimum inhibition concentration (MIC), and the minimum bactericidal concentration (MBC) values of CS-Tobramycin and CS-Amikacin particles crosslinked at 50%, 40%, and 20% mole ratios were also determined against *P. aeruginosa* as listed in Table 3. The zone diameter of 2.5 mg of CS-Tobramycin particles crosslinked at 50%, 40%, and 20% was found as 25 ± 1, 26 ± 1, and 23 ± 2 mm, respectively.

Similarly, MIC values of CS-Tobramycin particles crosslinked at 50%, 40%, and 20% was determined as 0.375, 0.375, and 0.750 mg/mL, respectively. It is obvious that antibacterial effects on the *P. aeruginosa* were totally dependent on the amount of release drug from the CS particle network and could be adjusted by changing the crosslinker degree from the particle network. Therefore, the highest antibacterial activity against *P. aeruginosa* was established in CS-Amikacin particles 40% with the lowest MIC and MBC value as 0.046 mg/mL and the broadest inhibition zone as 32 ± 2 mm with the highest drug release as 242 ± 4 μg/mg amikacin. Both formulations could be used in the treatment of *Pseudomonas* keratitis because of the almost similar release profiles and high antibacterial effects. In the literature, essential oil-loaded chitosan microcapsule embedded biodegradable sodium alginate/gelatin hydrogels were used to eliminate *P. aeruginosa,* but the antibacterial effect of the hydrogels was very weak with a high concentration of the MIC value at 39.3 mg/mL of cinnamon leaf oil as essential oil [52]. In another study, amikacin loaded to gelatin-coated poly(ethylene terephthalate) fibers were prepared and 15% of the loaded antibiotic were released within 7 days. The antibacterial activity of amikacin-loaded fibers was found effectively for 7 days, but *P. aeruginosa* was growing back after 10 days [53]. These studies indicated that the low MIC values and long-term inhibition abilities of CS-Tobramycin/Amikacin particles make them highly promising materials in the treatment of bacterial infection in ocular applications.

The blood compatibility of biomaterials is an important parameter for intravascular applications. The hemolysis ratio and blood clotting index of non-degradable CS particles and drug-loaded forms of CS-Tobramycin and CS-Amikacin were determined for various concentrations of CS-based particles from 100 to 1000 μg/mL. As can be seen in Figure 5a, CS particles were found as non-hemolytic materials with an acceptable 1.4 ± 0.2% hemolysis ratio even at a 1000 μg/mL concentration, whereas CS-Tobramycin and CS-Amikacin particles were found to possess slightly more hemolytic materials (in comparison to CS-Tobramycin) at a 1000 μg/mL concentration with 2.1 ± 0.1% and 2.0 ± 0.1% hemolysis ratio values.

Consequently, these particles were non-hemolytic at a 500 μg/mL concentration with 1.3 ± 0.2% and 1.7 ± 0.1% hemolysis ratio values, respectively. Similarly, CS particles were blood compatible with a high blood clotting index of 94.4 ± 1.7 at a 500 μg/mL concentration. The blood clotting index of CS-Tobramycin and CS-Amikacin particles decreased to 87.1 ± 0.9 and 84.4 ± 0.6 values for 500 μg/mL concentrations of the particles as seen in Figure 5b. These results show that hemo-compatible bare and drug-loaded CS particles can be used directly with up to a 500 μg/mL high concentration for safe intravascular applications. Some patients may require high doses of Tobramycin and Amikacin drugs for a long time to fight keratitis due to *P. aeruginosa* ulcers. The toxicity of these drugs in an overdose should be checked to avoid the side effects of the drugs. The cytotoxicity of tobramycin and amikacin drugs alone on L929 fibroblast cells were analyzed at different concentrations between 10 and 1000 μg/mL for 24 h incubation time as shown in Figure 6a. Cell viability with tobramycin was significantly decreased to 39 ± 11% at a 1000 μg/mL concentration from 96 ± 12% for a 10 μg/mL concentration of the drug. Similarly, amikacin has some toxicity at high concentrations with 49 ± 1% cell viability against fibroblasts at a 1000 μg/mL concentration (*p* < 0.0001 value).

The cytotoxicity of bare CS particles and drug-loaded CS-Tobramycin and CS-Amikacin particles was also determined by direct contact with L929 fibroblast cultures. The proliferation of the fibroblasts in the presence of CS-based particles at different concentrations was shown in Figure 6b. Even at high particle concentrations of 1000 μg/mL, the cell viability percentages were slightly decreased to 80 ± 5%, 71 ± 2%, and 71 ± 7% for CS particles, CS-Tobramycin particles, and CS-Amikacin particles, respectively. As represented in Figure 4, 1000 μg of drug-loaded CS particles could release more than 100 μg of drugs such as tobramycin or amikacin within 24 h. According to the cytotoxicity results, concentrations of 100 μg/mL or above of tobramycin or amikacin drugs destroyed the fibroblast cells with less than 60% cell viability. Previous studies reported that aminoglycosides such as tobramycin and amikacin have concentration-dependent nephrotoxicity and ototoxicity [41], and recommended the use of polymeric networks as carriers to decrease the toxicity of these drugs [25]. As reported in the literature, CS-derived polymeric materials have significant potential as a biomaterial for the treatment of many bactericidal diseases including corneal ulcers, and therefore their cytotoxicity was investigated. Han et al. found that selenium–chondroitin sulfate nanoparticles were slightly toxic with ~60% cell viability even at a low concentration of 340 ng/mL [54]. When compared with these CS-based materials, the drug-loaded CS-Tobramycin and CS-Amikacin particles were found to be much more biocompatible, with no significant cell loss up to a 250 μg/mL concentration with more than 80% cell viability values. It is clear that the use of the prepared CS particles as a drug carrier improves the bioavailability of the drugs and offer decreased toxicity as well as blood higher compatibility.

## 4. Conclusions

CS particles as drug carrying vehicles were prepared using DVS crosslinker at 50%, 40%, and 20% mole ratios per CS repeating unit. Distinctive degradation kinetics were obtained between the different crosslinker ratios and consequently 50%, 40%, and 20% CS particles had non-degradable, slightly degradable, and completely degradable structures under physiological conditions, respectively.

The non-degradable CS particles had a spherical shape and a smooth surface morphology with a size range of 0.5 to 5 μm, but slightly degradable CS particles were almost spherical and had a collapsed network with a 5–30 μm size range. These non-degradable CS particles were used as a carrier biomaterial to load tobramycin and amikacin antibiotics via the encapsulation technique for the treatment of *P. aeruginosa* keratitis. Significantly high amounts of these drugs, of approximately 250 μg/mg, were encapsulated within the CS particle network. These drug-loaded CS particles were promising biomaterials for the treatment of bacterial growth on the eye with long-term and sustainable drug release at efficient concentrations against *P. aeruginosa,* which cause ulcers on the cornea. Moreover, almost ideal blood compatibility and low cytotoxicity of the drug-loaded CS particles even at high concentrations of 1000 μg/mL providing a preventive effect for the toxicity of the drugs in biological systems were illustrated. Consequently, tobramycin- or amikacin-carrying CS particles could replace the currently employed treatment procedures for eye infections with their promising potentials, i.e., higher corneal residence time, higher and controllable drug loading ability, and lower toxicity with sustainable drug release kinetics from the particles that are derived from a natural polymer that is native to the eye. Hence, CS as a different antibiotic carrier for the treatment of bacterial infections in the various parts of the body, including the eyes, bestow a great alternative to currently used systems such as drops or pills. Moreover, CS particles can be used for the controlled and prolonged delivery of other drugs such as fungicides and anticancer drugs due to their non-toxic and blood compatibility nature.

## Figures and Tables

**Figure 1 pharmaceutics-14-01739-f001:**
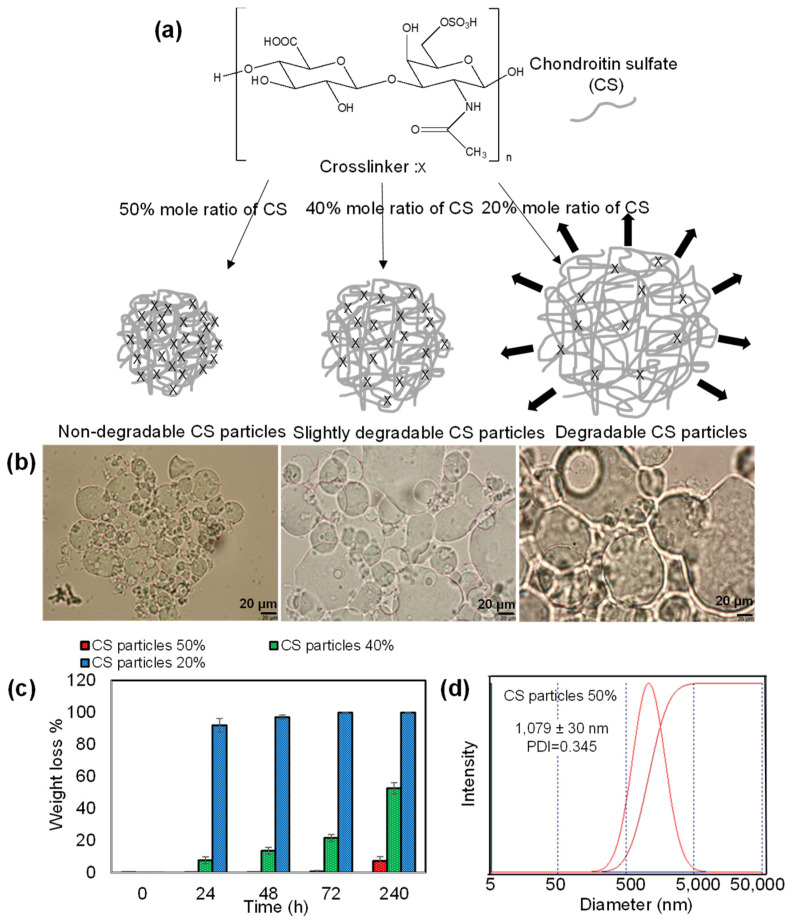
(**a**) Schematic representation of non-degradable, slightly degradable, and completely degradable CS particles prepared using three mole ratios of DVS crosslinker at 50, 40, and 20 mole% of CS repeating unit, and (**b**) their optical microscope images, and (**c**) weight loss% of CS particles prepared at 50, 40, and 20% crosslinked mole ratios were incubated for 24 h, 48 h, 72 h, and 240 h in physiological conditions at pH 7.4 and 37 °C. (**d**) Hydrodynamic size distribution of CS particles crosslinked at 50% mole ratio.

**Figure 2 pharmaceutics-14-01739-f002:**
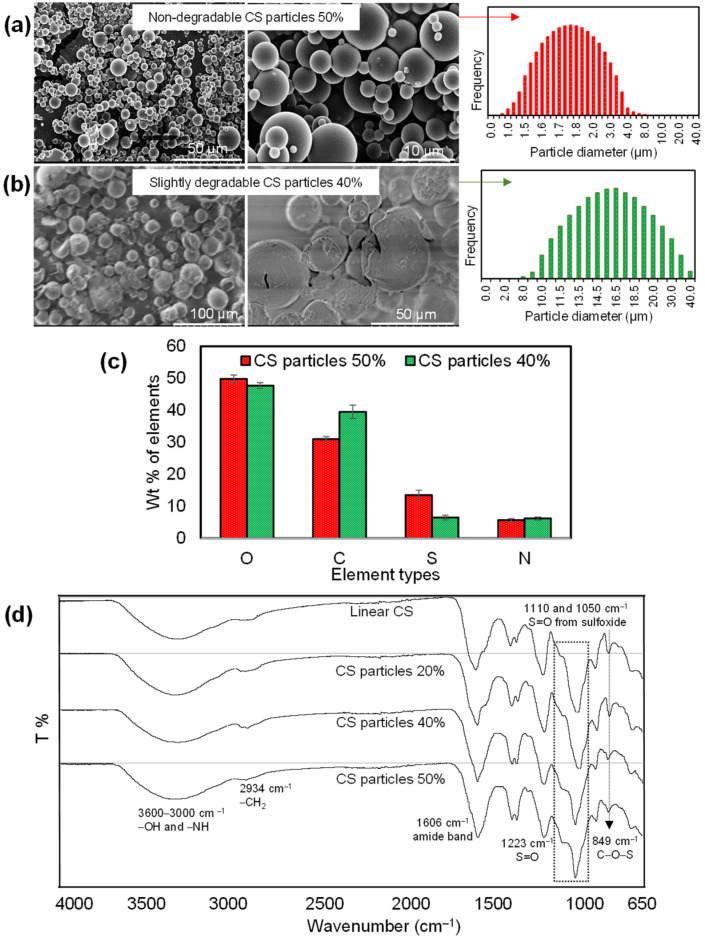
SEM images and size distribution of CS particles crosslinked at (**a**) 50%, (**b**) 40%, and (**c**) 20%. (**d**) FT-IR spectra of linear CS and its crosslinked forms, 20, 40, and 50% CS particles. (Size distribution was measured by ImageJ software according to SEM images.)

**Figure 3 pharmaceutics-14-01739-f003:**
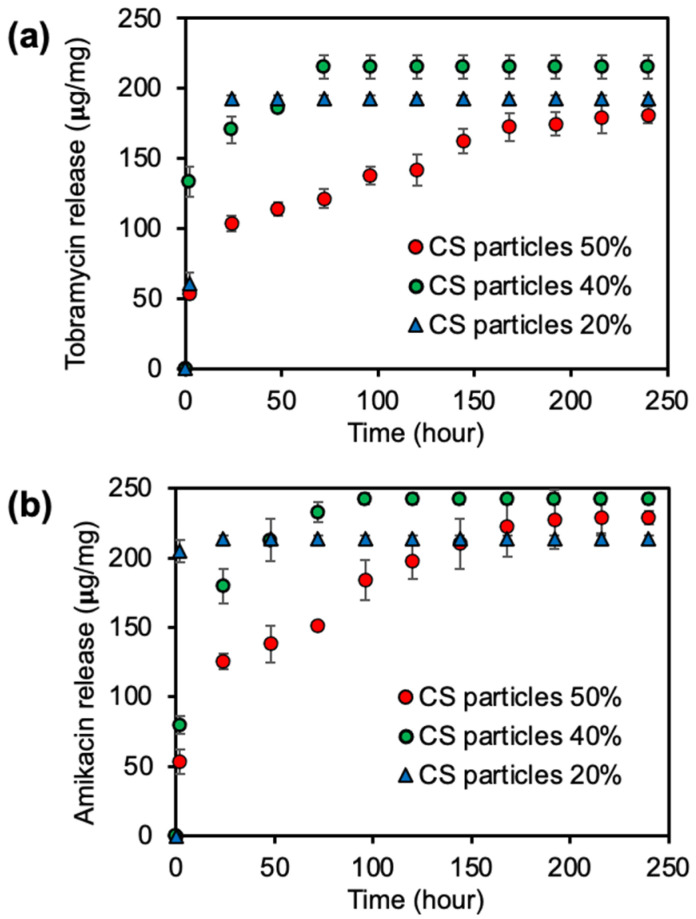
Drug release profiles of (**a**) CS-Tobramycin, and (**b**) CS-Amikacin particles crosslinked at 50, 40, and 20% mole ratios in pH 7.4 PBS at 37.5 °C.

**Figure 4 pharmaceutics-14-01739-f004:**
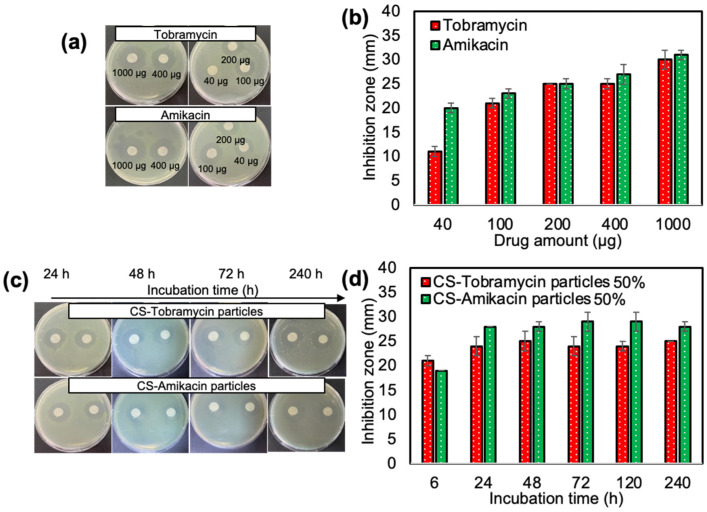
(**a**,**b**) Inhibition zone (mm) for tobramycin and amikacin drugs at different concentrations; 20 μL of 2–50 mg/mL for 24 h incubation, and (**c**,**d**) drug-loaded CS-Tobramycin 50% and CS-Amikacin particles 50% at 50 μL of 50 mg/mL concentration against *P. aeruginosa* at different incubation times.

**Figure 5 pharmaceutics-14-01739-f005:**
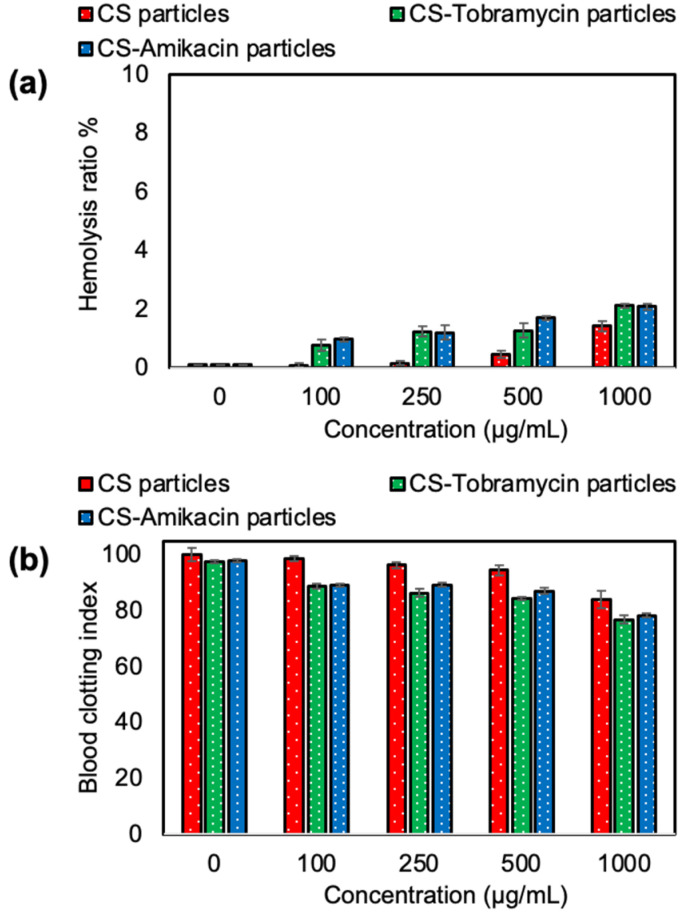
(**a**) Hemolysis and (**b**) blood clotting index of CS particles, CS-Tobramycin particles and CS-Amikacin particles crosslinked at 50% mole ratios.

**Figure 6 pharmaceutics-14-01739-f006:**
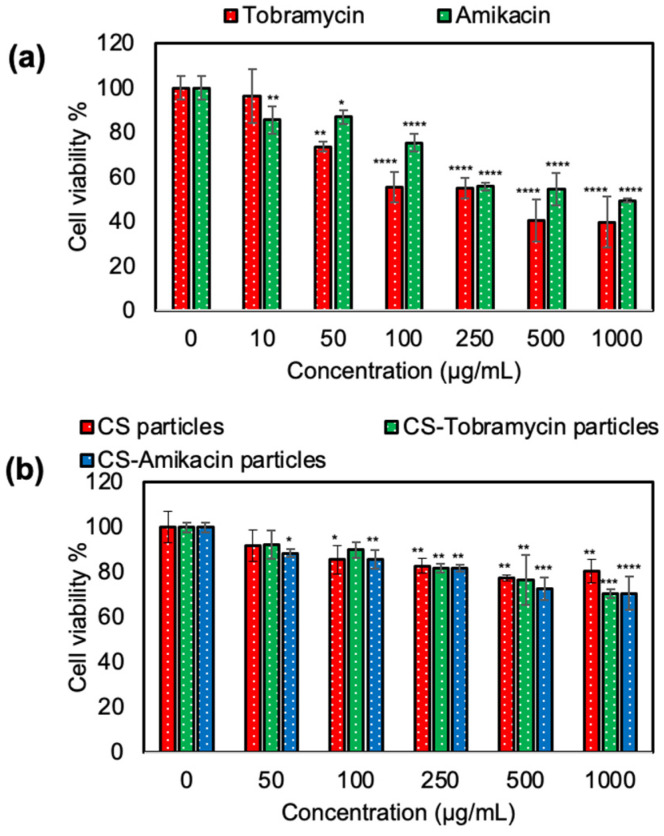
(**a**) Hemolysis and (**b**) blood clotting index of CS particles, CS-Tobramycin particles and CS-Amikacin particles crosslinked at 50% mole ratios. [*, **, ***, **** represent *p* < values of 0.05, 0.01, 0.001, and 0.0001, respectively].

**Table 1 pharmaceutics-14-01739-t001:** Hydrodynamic average size and polydispersity index values of CS, CS-Tobramycin, and CS-Amikacin particles crosslinked at 50% mole ratio.

Drug Carriers	Average Size (nm)	Polydispersity Index (PDI)
CS particles 50%	1079 ± 30	0.345
CS-Tobramycin particles 50%	830 ± 25	0.270
CS-Amikacin particles 50%	776 ± 57	0.519

**Table 2 pharmaceutics-14-01739-t002:** Maximum release amount (μg/mg) and release capacity (%) from CS-Tobramycin and CS-Amikacin particles crosslinked at 50%, 40%, and 20% mole ratios.

Drug Carriers	Release Amount (μg/mg)	Release Capacity (%)
CS-Tobramycin particles 50%	200 ± 2	80 ± 0.8
CS-Tobramycin particles 40%	215 ± 8	86 ± 3.2
CS-Tobramycin particles 20%	192 ± 3	76 ± 1.2
CS-Amikacin particles 50%	228 ± 5	91 ± 2.0
CS-Amikacin particles 40%	242 ± 4	96 ± 1.6
CS-Amikacin particles 20%	214 ± 2	80 ± 0.8

**Table 3 pharmaceutics-14-01739-t003:** Inhibition zone (mm), minimum inhibition concentration (MIC), and minimum bactericidal concentration (MBC) values of CS-Tobramycin and CS-Amikacin particles crosslinked at 50%, 40%, and 20% mole ratios against *P. aeruginosa*.

Drug Carriers	Inhibition Zone (mm)	MIC (mg/mL)	MIC (mg/mL)
CS-Tobramycin particles 50%	25 ± 1	0.375	1.500
CS-Tobramycin particles 40%	26 ± 1	0.375	0.750
CS-Tobramycin particles 20%	23 ± 2	0.750	1.500
CS-Amikacin particles 50%	28 ± 1	0.094	0.187
CS-Amikacin particles 40%	32 ± 2	0.046	0.046
CS-Amikacin particles 20%	26 ± 1	0.375	0.750

## Data Availability

The data presented in this study are available on request from the corresponding author.

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
