# Peer review of "Degradable and Non-Degradable Chondroitin Sulfate Particles with the Controlled Antibiotic Release for Bacterial Infections"

_pharmaceutics, 2022, doi:10.3390/pharmaceutics14081739_

Round 1
Reviewer 1 Report
The work done by S. Suner et al is novel and innovative, I will accept this manuscript for publication after revision.
1. Kindly add graphical abstract in the main manuscript as a schematic diagram to draw the reader's attention.
2. What does mean by Degradable and non-degradable chondroitin sulfate particles?
3. How CS particle is degraded?
4. It will be better if authors developed bacterial infection models and check antibiotic efficacy in vivo, also need to perform H&E staining.
5. Discussion part needs to revise and compare your results with already published literature to support your hypothesis.
6. Please revisit the entire manuscript for minor grammar and typo issues
Author Response
Reviewer 1
Comments and Suggestions for Authors
The work done by S. Suner et al is novel and innovative, I will accept this manuscript for publication after revision.
-We are grateful to the reviewer for the nice comments about the manuscript and we performed the following improvements as suggested;
- Kindly add graphical abstract in the main manuscript as a schematic diagram to draw the reader's attention.
-The images used for graphical abstract are also illustrated in Figure 1a and 2a. Although this would be a good idea, we the journal requires graphical abstract as a separate file.
- What does mean by Degradable and non-degradable chondroitin sulfate particles?
-Crosslinker network of polymeric particles that are CS particles can degrade in an aqueous solution including physiological saline solution due the swelling of the particles because of the functional hydrophilic groups such as -OH, -SO3H etc. The swelling and hence the degradation is affected by the pH of the medium, the ionic strength of the solution as well as the extent of the crosslinking of CS chains. The crosslinker extent or ratio used for the formation of CS particle afford tailoring degradable or non-degradable properties. As mentioned, on page 7, lines 328-335;
“In crosslinked polysaccharide particle preparation, DVS is commonly used, and depending on the functionality and hydrophilicity of the polysaccharide, the extent of DVS used in the crosslinked network formation can affect its hydrolytic degradability. In this study, CS was crosslinked with DVS at three different ratios, 50%, 40%, and 20% mole ratio of CS repeating unit to prepare non-degradable, slightly degradable, and degradable CS particles, respectively as illustrated in Figure 1a. To synthesize these particles, hydroxyl groups of CS were reacted with vinyl groups of DVS [23]. The ratio (degree) of crosslinker used in CS particle formation determines the degradability of CS particles [35].’’
- How CS particle is degraded?
-Hydrogels are non-soluble crosslinked polymeric networks and capable of swelling to a very high capacity in an aqueous solution depending on their chemical structure. The nature and the amount of crosslinker (or ratio) significantly affect the swelling ability e.g., the less the amount of crosslinker used in the hydrogel, microgel or particle (polymeric network) preparation, the higher the swelling degrees, and hence the effortless the hydrolytic degradation of the crosslinker networks by the time is attained. The higher the number of bonding between the hydroxyl groups of linear CS chains and the vinyl groups of DVS crosslinker, the tighter the crosslinked polymeric particles are obtained. Therefore, in this investigation, the crosslinker ratio <40% mole ratio of CS repeating unit rendered hydrolytically degradable CS particles in an aqueous solution or physiological solution as illustrated in Figure S1, due to the high swollen network that is also affected by the pH, ionic strength etc of the medium.
As mentioned, on page 7&8;
...“The CS particles containing a high ratio of crosslinker e.g., 50% mole ratio of DVS containing CS particles (based on the repeating unit of CS) were slightly swollen in DI water, and the crosslinked network did not degrade for a long time (up to 72h). As the crosslinking ratio was reduced e.g., to 40%, the CS particles were much more swollen than the 50% crosslinked CS particles, and relatively degradable CS particles were realized. Finally, CS particles containing a lower ratio of crosslinker e.g., 20% DVS crosslinked CS particle started to degrade within 5 min in an aqueous environment. Therefore, the 20% mole ratio of DVS is the limit for CS particle synthesis and the DVS extent that is <20% ratio results in no CS particle formation. Thus, the CS particle network with a low crosslinking ratio of %20 was extremely water swollen in an aqueous solution and could readily break down the crosslinked networks within a few minutes.”...
- It will be better if authors developed bacterial infection models and check antibiotic efficacy in vivo, also need to perform H&E staining.
-The aim of this study is to design tobramycin and amikacin carrier materials with a controllable release capability of these antibiotic drugs using degradable CS particles. The release kinetic and their in vitro antibacterial effects against Pseudomonas aeruginosa were also investigated and show the potential of these carriers in the treatment of the Pseudomonas keratitis. The determination of the in vivo antibacterial activities of these carriers is the next step and will be investigated in our upcoming research.
- Discussion part needs to revise and compare your results with already published literature to support your hypothesis.
-Some new references were added in the results and discussion section to improve the discussion part of manuscript for drug release results on pp 12, and for antibacterial activity of particles on pp 14, respectively, as;
... “In the literature, tobramycin/amikacin carriers such as alginate/chitosan particles [47], polyethylene glycol based hydrogel [48], and liposomal systems [49] were designed to prevent infections caused by P. aeruginosa. Deacon et al., reported that algi-nate/chitosan based particles were loaded with 92±18 g/mg tobramycin and 80% of the loaded tobramycin could be delivered from the polymeric network within 48 h [47]. Postic et al., stated that polyethylene glycolbased hydrogels could release about 40 g/mL of amikacin in 5 h [48]. In a study, amikacin loaded nanoparticles from Eu-dragit RS 100/Eudragit RL 100 polymer composition were used in the bacterial treat-ment of ocular infection. The release profiles from the nanocarrier system were finished within 12 h with about 90.8% release capacity [50]. Therefore, in comparison to these carrier systems, CS-Tobramycin/Amikacin particles reported here is promising biomaterial due to the higher antibiotic loading ability and longer time antibiotic re-lease capacities.”
...” In the literature, essential oil loaded chitosan microcapsule embedded biodegradable sodium alginate/gelatin hydrogels were used to eliminate P. aeruginosa, but the anti-bacterial effect of the hydrogels was very weak with a high concentration of MIC value at 39.3 mg/mL of cinnamon leaf oil as essential oil [52]. In another study, amikacin loaded to gelatin coated poly(ethylene terephthalate) fibers were prepared and 15% of the loaded antibiotic were released within 7 days. The antibacterial activity of amika-cin loaded fibers was found effectively for 7 days, but P. aeruginosa was growing back after 10 days [53]. These studies indicated that low MIC values and long-term inhibi-tion abilities of CS-Tobramycin/Amikacin particles make them highly promising mate-rials in the treatment of bacterial infection in ocular applications.”...
- Please revisit the entire manuscript for minor grammar and typo issues.
-The manuscript was proofread and edited by a native speaker of English. The editing certificate was added.

Reviewer 2 Report
The authors submitted manuscript regarding degradation and non-degradation behaviour and antibacterial activity of the chondroitin sulphate based nanoparticles is quite interesting and useful.
As the chondroitin sulphate is a polysaccharide based and shows some antibacterial activities. BY using and exploring with the use of Tobramycin and Amikacin in CS nanoparticles is useful and interesting.
Some major findings were observed and need to clarified.
1. Abbreviation to be kept uniform first came and expand first.
2. Is there any specific reason to select these two antibiotic Tobramycin and Amikacin for antibacterial activities.
3. Fig 2 shows the non-degradable 50% and slightly degradable CS particles 40% and author did not shows the degradable CS particle SEM image and PS distribution.
4.They havnt check and provided the stability data with PS distribution and SEM image at different time point, which would be more useful in dragdation or non-degradation.
5.Table 1 PS data is very wide and need to clarify as their SD values is much higher side.
6.There is no trend was observing from release amount of tobramycin from 50%, 40% and 20% mole ratios. Additionally, is we are going to develop as a long-acting formulation based on the release studies.
7. I think authors should add some more bacteria species instead only Pseudomonas sps.
In my opinion, author revise this manuscript with the above mentioned suggestion as it is premature for publication in this current format.
Author Response
Reviewer 2
Comments and Suggestions for Authors
The authors submitted manuscript regarding degradation and non-degradation behaviour and antibacterial activity of the chondroitin sulphate based nanoparticles is quite interesting and useful.
As the chondroitin sulphate is a polysaccharide based and shows some antibacterial activities. BY using and exploring with the use of Tobramycin and Amikacin in CS nanoparticles is useful and interesting.
-We are thankful for the nice comment about manuscript.
Some major findings were observed and need to clarified.
- Abbreviation to be kept uniform first came and expand first.
-Thank you, the required corrections were done.
- Is there any specific reason to select these two antibiotic Tobramycin and Amikacin for antibacterial activities.
These antibiotics are specific for Pseudomonas aeruginosa keratitis and is commonly used antibiotics in the clinical application as mentioned on pp 2 of the revised manuscript as;
“In the treatment of bacterial ulcers on the cornea such as Pseudomonas keratitis, the high ophthalmic toxicity and poor pharmacokinetics of the common drugs such as tobramycin and amikacin offer limited use in the clinical application due to the low drug permeability to the epithelial membrane necessitating frequent administration [24].”
- Fig 2 shows the non-degradable 50% and slightly degradable CS particles 40% and author did not shows the degradable CS particle SEM image and PS distribution.
-SEM image and PS distribution of degradable CS particles 20% was also added in Figure 2c. The results were explained on pp 9 of the revised manuscript as;
...“On the other hand, the surface structure of the dry CS particles crosslinked with a 20% mole ratio is rough with almost spherical shapes with particle sizes ranging from 5 to 50 m. The size distribution range for 50% crosslinked CS particles were found to de-crease almost ten-fold and five-fold with respect to 40% and 20% crosslinked CS parti-cles, respectively..”...
4.They havnt check and provided the stability data with PS distribution and SEM image at different time point, which would be more useful in dragdation or non-degradation.
-CS particles were hydrolytically degradable as given in Supplementary Figure 1 according to optic microscope images, but not degradable at dry forms which visualized by SEM images. Furthermore, particle size distribution was determined using ImageJ software from the SEM images of CS particles as given in Figures 2a and 2b. As the particle degradability will be used for drug delivery, their degradability afforded controlled drug release it is not that significant to visualize the particles’ SEM images at different time point.
5.Table 1 PS data is very wide and need to clarify as their SD values is much higher side.
-The hydrodynamic average diameter by using DLS measurements could be carried out only for 50% mole ratio crosslinked CS particles and its drug loaded forms because of the detection limit of DLS which is maximum only a few mm (maybe Max 5mm) particles size. The hydrodynamic size distribution graphic of CS particles crosslinked at 50% mole ratio and the measurement results with polydispersity index were given in Figure 1d and Table 1. It could be said that highly crosslinked CS particles e.g., at 50% mole ratio and its drug loaded forms do not show very wide size distribution in comparison with the others lesser amount of crosslinker CS particles which has polydispersity index values of 0.345 and 0.519. for 40 and 20% mole ratio of DVS crosslinked CS particles.
6.There is no trend was observing from release amount of tobramycin from 50%, 40% and 20% mole ratios. Additionally, is we are going to develop as a long-acting formulation based on the release studies.
-Actually, 40% and 20% mole ratios of drug loaded CS particles were shown almost similar release profiles, but 50% mole ratio of drug loaded CS particles provide sustainable and long-acting release profiles in about 200 h with linear drug releasing capacity.
- I think authors should add some more bacteria species instead only Pseudomonas sps.
The aim of this study was to investigate the potential usability of CS particles as antibiotic carrier for potential treatment of Pseudomonas keratitis as some bacterial ulcers on the cornea. Therefore, antibacterial effects of tobramycin/amikacin loaded CS particles against different bacteria species was not necessary for this research.
In my opinion, author revise this manuscript with the above mentioned suggestion as it is premature for publication in this current format.
-Done as suggested, and we are thankful to the review for the constructive suggestions that we believe help to improve the quality of the manuscript greatly.

Round 2
Reviewer 1 Report
Accepted in present form